# Foliar Application of *Rhodopseudomonas palustris* Enhances the Rice Crop Growth and Yield under Field Conditions

**DOI:** 10.3390/plants11192452

**Published:** 2022-09-20

**Authors:** Kuei Shan Yen, Laurence Shiva Sundar, Yun-Yang Chao

**Affiliations:** 1Department of Plant Industry, National Pingtung University of Science and Technology, 1, Shuefu Road, Neipu, Pingtung 912301, Taiwan; b10911043@mail.npust.edu.tw; 2Department of Tropical Agriculture and International Cooperation, National Pingtung University of Science and Technology, 1, Shuefu Road, Neipu, Pingtung 912301, Taiwan; laurenceshivasundar@gmail.com; 3Department of Crop Science, College of Agriculture, Fisheries and Forestry, Koronivia Campus, Fiji National University, Nausori P.O. Box 1544, Fiji

**Keywords:** anthropogenic activities, antioxidant enzymes, cereal crop, climate change, crop production, crop yield, harvest index, land degradation, plant growth-promoting bacteria

## Abstract

Anthropogenic activities causing climate change and other environmental effects are lowering crop yield by deteriorating the growing environment for crops. Rice, a globally important cereal crop, is under production threat due to climate change and land degradation. This research aims to sustainably improve rice growth and yield by using *Rhodopseudomonas palustris*, a plant growth-promoting bacteria that has recently gained much attention in crop production. The experiment was set up in two fields, one as a control and the other as a PNSB-treated field. The foliar application of treatment was made fortnightly until the end of the vegetative stage. Data on the growth, yield, and antioxidant enzymes were collected weekly. The results of this experiment indicate no significant differences in the plant height, root volume, average grain per panicle, biological yield, grain fertility, and antioxidant enzyme activity between the PNSB-treated and untreated plants. However, a significant increase in the tiller number, leaf chlorophyll content and lodging resistance were noted with PNSB treatment. Likewise, PNSB-treatment significantly increased root length (25%), root dry weight (57%), productive tillers per plants (26%), average grains per plant (38%), grain yield (33%), 1000 grain weight (1.6%), and harvest index (41%). Hence, from this research, it can be concluded that foliar application of PNSB on rice crops under field conditions improves crop growth and yield, although it does not affect antioxidant enzyme activity.

## 1. Introduction

The world population and people’s changing diets are driving higher demand for food now and in the future. Therefore, farmers need to increase food production yearly with the available resources to meet this demand. Currently, the production of crops is enhanced with an expansion in the cultivation area with the higher investment made in the marginal lands [1]. However, this investment in the future will be complex due to reduced water availability caused by groundwater depletion and the competition for natural resources [2]. Therefore, more crops need to be produced in a smaller cultivation area with a reduced amount of water [3].

On the other hand, the production of crops in the future will be stressful as climate change is expected to affect crop yields negatively [4]. Increased carbon dioxide (CO_2_) for photosynthesis and increased temperatures in temperate regions due to climate change may favor plant growth. However, the increasing temperature in non-temperate zones, intensity and frequency of precipitation, and extreme weather events will directly affect agricultural productivity [4]. The increasing temperature will result in lower crop yield while encouraging pests and weed proliferation [5], even though it will lead to the extinction of certain pests and pathogens that used to thrive best in lower temperatures. At the same time, changes in the intensity and frequency of precipitation will result in short-run crop failures and long-term production decline [5].

Although agriculture is highly affected by climate change impacts, it is also known to be one of the significant contributors to the production and release of greenhouse gas. Methane (CH_4_), the second most important greenhouse gas, has the highest warming potential than CO_2_, the most important greenhouse gas. Paddy fields are a common source of anthropogenic CH_4_ emissions, accounting for around 10–20% of emissions [6]. Reducing rice production could be a solution to lowering the CH_4_ emission; however, it is impossible as rice is among the most important staple food for nearly 50 percent of the world population [7,8,9,10], including those living in developed and developing countries. Therefore, alternative methods are needed to enhance rice crop productivity while lowering its negative environmental impact.

Microorganisms, mainly bacteria, play a vital role in sustainable agriculture with the ability to promote plant growth, reduce plant stress caused by abiotic and biotic factors, nutrient recycling, and manage soil fertility, leading to low usage of chemical fertilizers and pesticides that pose adverse risks to soil or crops [11]. Purple non-sulfur bacteria (PNSB) are phototropic microorganisms that can enhance plant growth, boost resistance to environmental stress, improve the yield and quality of edible parts, alleviate salinity stress, improve plants’ resistance to heavy metal stress, and mitigate greenhouse gas emissions [12]. The *Rhodopseudomonas palustris* species of PNSB are among the most important bacteria widely used in agriculture to promote plant growth and yield. This bacteria species can grow under all primary metabolisms (photo-autotrophy, photo-heterotrophy, chemo-autotrophy, and chemo-heterotrophy) and in aerobic and anaerobic conditions [13]. Crops such as pak choi [14,15,16], stevia [17], tobacco [18,19], mushroom [20], Chinese dwarf cherry [21], bean [22], and rice [23,24,25,26] have been inoculated with *Rhodopseudomonas palustris,* and positive results were obtained.

Likewise, several other studies to investigate the beneficial effects of PNSB on rice crop plants were carried out, and most of them have reported positive results. However, these investigations were done under different growing environments, application methods, and application rates and investigated different parameters under different treatments with different PNSB species that did not fully exploit the beneficial effects of the PNSB on rice crops. Therefore, this research was designed to investigate the beneficial effects of foliar application of PNSB on rice crop plants under the field conditions of a tropical climate.

## 2. Results

### 2.1. Agrometeorological Data

The experiment was conducted in an open field, and important agrometeorological data such as temperature, relative humidity, and light intensity were recorded. Throughout the growing period, the temperature increased gradually, as shown in Figure 1a. The temperature remained very low for the first 6 WAT, gradually increasing over the growing period. The highest weekly average temperature of 31.9 °C was recorded on WAT 20, whereas the lowest temperature of 19.6 °C was recorded on WAT 6 (Figure 1a). On the other hand, the relative humidity (RH) gradually lowered throughout the growing period, especially until WAT 15, with an increase in the growing temperature, as shown in Figure 1a. The highest average weekly RH of 85.2% was recorded at WAT 2, whereas the lowest RH of 63.8% was recorded at WAT 12, as shown in Figure 1a. A slight increase in RH with a temperature increase could be due to the precipitation during these periods, as indicated by the low light intensity recorded. The agrometeorological data show that the light intensity (LI) fluctuated throughout the growing period (Figure 1b). The highest LI of 1068 µmol m^−2^ s^−1^ was recorded at WAT 8, whereas the lowest LI of 343 µmol m^−2^ s^−1^ was recorded at WAT 17, as shown in Figure 1b.

### 2.2. Agronomic Performances

Field performances of the rice crop were evaluated based on the plant height, tiller number, leaf chlorophyll content, and lodging resistance. A smooth incline in the plant height for both control and PNSB-treated plants was observed, as shown in Figure 2a. At WAT 12, during the final data collection, the result indicates that the PNSB-treated plants had the highest plant height of 99.3 cm compared to the control with just 95.4 cm. A similar trend was observed throughout the growing period, where the PNSB-treated plants had the highest plant height compared to the untreated plants (Figure 2a). However, the independent sample *t*-test results indicate no significant differences in the average plant height between the PNSB-treated (68.2 cm) and untreated plants (63.9 cm).

As for the tiller number, a slick sigmoidal shape was observed throughout the growing period, clearly evident in the results for the PNSB-treated plants, as shown in Figure 2b. The results show a gradual increase in the tiller number from WAT 4 to WAT 9, after which a decline in the tiller number was observed. The maximum tiller number was observed at WAT 9, with PNSB-treated plants having the highest tiller number of 23.4 compared to untreated plants with 20.6 average tillers (Figure 2b). The claim was also in line with the analysis performed by the independent sample *t*-test, indicating that the PNSB-treated plants had significantly higher average tiller numbers (19.1) than the untreated plants (16.7).

Moreover, the leaf chlorophyll content, measured in the SPAD unit, was also investigated. The results show a similar trend for both PNSB-treated and untreated plants (Figure 2c). After WAT 10, close to the reproductive stage, the leaf chlorophyll content declined for both PNSB-treated and untreated plants; however, throughout the growing period, the leaf chlorophyll content of the PNSB-treated rice crop plants was higher than the untreated plants. The highest chlorophyll content of PNSB-treated plants was at WAT 6 (36.7 SPAD value) and for untreated was at WAT 10 (35.3 SPAD value), as shown in Figure 2c. The independent sample *t*-test analysis reveals that the PNSB-treated plants had significantly higher average leaf chlorophyll content (34.9 SPAD value) compared to untreated plants (33.7 SPAD value).

Similarly, the plant lodging resistance had a smooth increase over the growing period, with PNSB-treated plants having a continuous increase whereas the lodging resistance of the untreated plants plunged from WAT 11, as shown in Figure 2d. The PNSB-treated plants had higher lodging resistance throughout the growing period, with the highest at WAT 12 (187 kPa) compared to the untreated plants, with the highest at WAT 11 (145 kPa), as shown in Figure 2d. The independent sample *t*-test analysis indicates that the PNSB-treated plants had a significantly higher average plant lodging resistance of 100 kPa, compared to untreated plants with 72.8 kPa.

### 2.3. Rice Root Growth and Biomass

Other than above ground, the below-ground performances of rice crop plants under PNSB-treated and untreated were evaluated. Root parameters such as root length, volume, and dry weight were investigated as part of this research. The results show that the rice root length (shown in Figure 3) and dry mass significantly increased with PNSB treatment, as shown in Table 1. However, as for the root volume, even though the result shows that PNSB-treated plants had the highest root volume, the independent sample *t*-test analysis shows no significant differences between the PNSB-treated and untreated plants.

### 2.4. Antioxidant Enzyme Activity

The antioxidant enzyme activity of rice crop plants was analyzed to understand the antioxidant capacity of the PNSB-treated and untreated plants. For the APX activity, at WAT 5 and WAT 6 (early stage of plant growth), the result shows that the PNSB-treated plants had significantly higher APX activity compared to untreated plants, whereas at WAT 8 and WAT 12 (mid and late stage of plant growth, respectively), the untreated plants had significantly higher APX activity (Figure 4a). However, the overall results determined by the independent sample *t*-test indicated that there were no significant differences in the APX activity between the PNSB-treated and untreated plants.

Slightly similar results were obtained for the GR activity, showing that at WAT 8 (mid-stage of plant growth), the untreated plant had significantly highest GR activity compared to the PNSB-treated plants (Figure 4b). However, the overall results determined by the independent sample *t*-test indicated that there were no significant differences in the GR activity between the PNSB-treated and untreated plants.

On the other hand, the CAT and SOD activity of rice crop plants shows similar results. Throughout the growing period, fluctuations in the CAT and SOD activity were observed between the PNSB-treated and untreated plants, but no significant differences were noted. The independent sample *t*-test also confirms that there were no significant differences in the CAT and SOD activity between the PNSB-treated and untreated plants, as shown in Figure 4c,d, respectively.

### 2.5. Yield and Yield Components

Yield, a measure of crops produced per unit area, is an important factor that enables farmers to invest in certain crops. The higher the crop yield, the more the demand for its cultivation. In order to understand the rice crop yield, the components such as productive tillers per plant, average grains per plant, average grains per panicle, and biological yield were evaluated. The results show that PNSB treatment significantly improved the tillers per plant from 48% to 64% and the average grains per plant from 1614 to 2584, as shown in Table 2. As for the average grains per panicle, no significant differences were observed for the PNSB-treated plants (82) and untreated plants (81). Similarly, for the biological yield, no significant differences were observed for the PNSB-treated plants (14 t ha^−1^, respectively) and untreated plants (18 t ha^−1^). Therefore, the yield component results indicate that the PNSB-treated plants will be dominant in yield over the untreated plants.

Moreover, to understand the yield performances of the rice crop treated with PNSB, the yield parameters such as the grain yield, grain fertility, 1000 grain weight, and harvest index were evaluated. The result shows that treating with PNSB significantly increased the grain yield from 5.42 t ha^−1^ to 8.10 t ha^−1^, as shown in Table 3.

However, grain fertility shows no significant differences between PNSB-treated (46.99%) and untreated (42.02%) rice crop plants. As for the 1000 grain weight, the results show that treating with PNSB significantly increases the 1000 grain weight from 23.98 g to 24.37 g, as shown in Table 3. Similarly, PNSB treatment significantly enhanced the harvest index from 0.33% for control to 0.56% for PNSB-treated plants (Table 3).

## 3. Discussion

Like other crops, rice also has an optimum temperature for successful growth and development. The optimum temperature for rice cultivation is between 25 °C to 35 °C [27]. An increase or decrease in the temperature other than the optimum temperature will cause an alteration in physiological activity and can also lead to a different development pathway [28]. Changes in the optimum temperature of rice crop plants can lead to a slower developmental rate and increase the time for reaching the heading stage [29]. The reproductive stage in rice crops is more sensitive than the vegetative stage [30]; however, an extreme temperature at the vegetative stage has no impact on the reproductive stage [31]. The growing environment temperature in this experiment was between 19.6 °C to 31.9 °C. From WAT 1-6, 8, and 11, the growing environment temperature was below 25 °C, which is lower than the optimum temperature for rice crop growth and development. At the late stage of the plant growth, the temperature stayed lower to around 31.9 °C compared to the 35 °C optimum temperature. Therefore, even though the temperature increased over the growing period, it was relatively lower than the optimum temperature for growing rice crops.

In addition to the temperature, rice crop growth is also determined by the light intensity. Rice grows best with 12 h of light, with light intensity between 500–1000 µmol m^−2^ s^−1^ [32]. Therefore, the main characteristics of rice crop development are determined by light intensity, among the most significant environmental factors. Continuous rainfall or cloudy weather can result in significant yield loss and poor grain quality, especially during the grain filling stage [33]. In this experiment, the light intensity was between 343 µmol m^−2^ s^−1^ to 1068 µmol m^−2^ s^−1^. At WAT 1, 5, 11, and 16-18, the light intensity was lower than the optimum light intensity required for successful growth and yield production. The light intensity at the grain filling stage was also noted to be relatively lower than the optimum light intensity, which could have resulted in a lower grain yield. Overall, the results show that the light intensity was relatively lower throughout the growing period, especially at the later stage of plant growth, when the demand for light by plants is much higher, specifically at the grain filling stage.

The cooler temperature and lower light, especially at the grain filling stage, substantially affected the grain fertility in both PNSB-treated and untreated plants, as shown in Table 3. In addition to rice, the lower light intensity also affected other crops and fruits in the country in the same growing period, such as millet, scallions, onions, lychees, and mangoes [34]. However, in this experiment, even under cooler temperatures and lower light intensity, the effectiveness of PNSB in yield improvement was still evident. The results of this experiment show that PNSB treatment significantly improved the productive tillers per plant, average grains per plant, grain yield, 1000-grain weight, and, therefore, the harvest index. Rice is grown in Taiwan twice a year, with the first cropping season from February to June (late winter to mid-summer) and the second cropping season from July to November (mid-summer to early winter) [35]. Due to differences in the temperature throughout the growing season, the first crop takes 128 days to harvest, whereas the second takes 115 days to harvest [36]. Therefore, the yield of the first season crop is higher (6.18 t ha^−1^) than the second one (4.31 t ha^−1^) [36]. In this research, the rice was grown in the first cropping season, and the result shows that the PNSB treatment increased the grain yield by 25% (8.10 t ha^−1^) from the average grain yield of this variety. On the other hand, the untreated plant had a grain yield of 5.42 t ha^−1^ (lower than first-season cropping but higher than second season cropping), which was 12% lower than this variety’s average grain yield due to slight cooler temperature and lower light intensity.

Likewise, Kantachote and colleagues [25] reported that PNSB treatment significantly improved the grain yield under field conditions using the same species of PNSB. The pot experiment by Harada et al. [23] using the same PNSB species also shows that PNSB treatment has significantly increased the grain yield of the rice crop. Even using different species of PNSB (*R. capsulatus*), Yoshida and the team [37] reported a significant increase in grain yield with PNSB treatment under field conditions. Elbadry et al. [38] reported that treating *Rhodobacter capsulatus* species of PNSB significantly increased the grain yield of rice crop plants. Finally, the experiment results of Gamal-Eldin and Elbanna [39] using *R. capsulatus* indicated that the number of productive tillers and the grain yield increased with PNSB treatment; however, the experiment was carried out with different application methods and rates. As for the average grains per panicle and biological yield, no significant differences were noted under PNSB treatment. However, previous studies have shown that PNSB treatment does improve the grain per panicle and the biological yield of the rice crop plants [24,25,26,38,40,41], which could be due to different growing conditions, application methods, and application rates used in different experiments.

Furthermore, the PNSB treatment also significantly affected the growth of rice crop plants. Apart from the plant height, the tiller number, leaf chlorophyll content, and lodging, resistance showed significant improvement with PNSB treatment. The plant height results showed no significant differences between the PNSB-treated and untreated plants. Similar results were obtained by Kantachote et al. [25], Kantha et al. [24], and Nookongbut et al. [26] using the same species of PNSB (*R. palustris*) but under different growing conditions, application methods, and application rates. Likewise, similar results were obtained by Gamal-Eldin and Elbanna [41] using *R. capsulatus* under different growing conditions, application methods, and rates. The plant height result for this experiment was not similar to that obtained by Harada et al. [23] using the same species of PNSB, which indicated that PNSB treatment significantly increased the plant height. However, the experiment was conducted under different growing conditions, application methods, and rates.

Moreover, the tiller number, one of the most crucial factors determining the grain yield [42,43,44], was also investigated in this research. The tiller number in both treated and untreated plants increased over the growing period and reached the maximum tiller number (23.4 for PNSB-treated plants and 20.6 for untreated plants) at WAT 9, after which the tiller number was reduced. The tiller number for the Kaohsiung 147 variety, an improved variety used in this research, shows similar findings to Pawar et al. [45] that indicate that modern rice varieties produce 20–25 tillers, including primary, secondary, and tertiary tillers. The reduction in the tiller number after the maximum tiller numbers was due to the competition effects, where the portion of the late tillers will die off after the maximum tiller number has been reached [46]. The results of this experiment show that PNSB-treated plants had significantly higher tiller numbers than untreated plants. No comparative investigation was done to evaluate the tiller number performances of rice crop plants with PNSB treatment.

As for the leaf chlorophyll content, the results of this experiment for the PNSB-treated rice crop plants were significantly higher than the untreated plants. Similar results were obtained by Nunkaew et al. [47] using the same species of PNSB under different growing conditions, application methods, and rates. However, Nookongbut et al. [26] concluded that PNSB treatment does not affect the leaf chlorophyll content of rice crops using the same PNSB species, but the research was done under different growing conditions, application methods, and rates.

Similarly, the plant lodging resistance results in this experiment show that the PNSB-treated plants had a significantly higher lodging resistance than the untreated plants. The higher lodging resistance under PNSB treatment could be due to a higher number of roots [40,48] and longer roots [24,48] initiated by PNSB inoculation. This claim was overturned by the research conducted by Elbadry and Elbanna [40], stating that the PNSB inoculation decreases the root length, whereas Nookongbut and colleagues concluded that PNSB treatment has no significant effect on the root length. However, after investigating the below-ground performances of the rice crop in this research, the results agree that PNSB treatment positively affects the root parameters. The results show that PNSB treatment on rice crops significantly increases root length and dry weight, which explains a higher lodging resistance in PNSB-treated plants. A higher root volume was also observed with PNSB treatment, but the independent sample *t*-test indicates that it was not significantly higher than the untreated plants.

On the other hand, the antioxidant enzyme (APX, CAT, GR, and SOD) activity results showed no significant differences among the PNSB-treated and untreated plants. The Kaohsiung 147 rice crop used in this research was developed to be grown in a tropical climate with higher light intensity and, as such, is not a low light tolerant variety. During low light, the antioxidative and osmotic regulating systems must have failed due to the low light intolerant ability, resulting in cell membrane damage [33]. The other studies also demonstrated a similar theory by analyzing the related parameters [49,50,51]. This theory could be one reason the PNSB application did not enhance the antioxidant enzyme activity in rice crop plants. Another reason could be that the strain of PNSB selected for this experiment was not very effective in enhancing antioxidant enzyme activity, which was revealed in the study by Nunkaew et al. [47]. The findings revealed that only the PP803 strain of *R. palustris* significantly affected the antioxidant enzyme activity of rice, outperforming the TN114 strain. Similarly, the result of this experiment was also parallel to the finding by Nookongbut et al. [26], revealing that even under stress conditions (As stress), the PNSB has no significant effect on the antioxidant enzyme activity of the rice crop plants. Therefore, further experiments under field conditions are needed to understand the effectiveness of PNSB in enhancing the antioxidant enzyme activity of rice and other field crops.

## 4. Materials and Methods

### 4.1. Experimental Location and Setup

The current study was carried out in the practice farm of the Department of Plant Industry, National Pingtung University of Science and Technology (NPUST), Taiwan, R.O.C, from January 2022 to June 2022 (the main rice growing season in Taiwan). The farm is located in an open area with coordinates of 22°38′54.0″ N and 120°37′01.9″ E. Two separate fields, each measuring around 21 m in length and 9 m in width, were selected as the research area. One of the fields was used as a control field where no treatment was applied, whereas the other was used as a treatment field. The blocks were set up parallel to each other in the same area under similar environmental conditions. All management practices, from land preparation to harvesting, were carried out similarly in each block to avoid biases in the results. In each field, Kaohsiung 147 variety of rice crop was transplanted at the 3-leaf stage using the rice transplanter. At 4 weeks after transplanting (WAT), 10 plants in each field were tagged for field data collection, whereas for the analysis of antioxidant enzyme activity, random plants were selected weekly for sample collection.

Weather conditions such as air temperature, relative humidity, and light intensity were monitored using the fully automated KLIMALOG Microclimate Environment Monitoring System by Taiwan Hibot Co., Ltd., Kaohsiung, Taiwan R.O.C. The data was extracted from the online system weekly and recorded to make decisions on management practices.

### 4.2. Bacteria Preparation and Application

The *Rhodopseudomonas palustris* species of purple non-sulfur bacteria (PNSB) was selected for this research due to its numerous benefits to rice crops when grown under different environmental conditions, as shown by various studies [23,24,25,26]. The PNSB was prepared using the original stock purchased from Dawushan Cultural and Education Foundation, Taiwan R.O.C. The culture medium was prepared according to Lee et al. [52] with a few modifications based on the available materials and as suggested by Dawushan Cultural and Education Foundation. The bacteria were cultured in a 20 L transparent water bottle and left in the greenhouse (under indirect sunlight) for two weeks for the bacteria to be fully grown, as indicated by the dark maroon color. The bottle was checked and shaken daily for evenly spread of the prepared culture media throughout the bottle for smooth and even bacteria growth. After the medium changed its color to dark maroon within 14 days of culture, 10 mL of the stock solution was taken for laboratory analysis to determine the colony forming unit (CFU). The original CFU of 2.24 × 10^7^ was determined by the standard plate count (SPC) technique, which was then diluted and adjusted to 2.46 x 10^8^ to meet the research needs. At four WAT, the rice crop plants were inoculated in the field with PNSB as a foliar application, which was repeated fortnightly until the early reproductive stage.

### 4.3. Crop Management Practices

Crop management practices such as fertilizer application, irrigation frequency, weeding, and pest and disease management were done as and when needed. During the dry field preparation, farmyard manure (N-2.6, P-1.9, K-1.4) was used as a basal application. During plant growth, the “Heiwangte No. 43” compound fertilizer 15-15-15-3(MgO) 50 (O.M.) fertilizer, supplied by Taiwan Fertilizer Co., Ltd. (Taipei, Taiwan) was applied in 3 splits, i.e., at the early leaf development stage, mid-tillering stage, and early reproductive stage, as suggested by the Miaoli District Agricultural Research and Extension Station, Miaoli County, Taiwan, R.O.C.

The rice crop field was irrigated using the wet and dry technique, where the field was irrigated for 24 h and then left for 3 days to dry off. This way, the plant roots can absorb enough oxygen to carry out respiration, and at the same time, the weed and algae growth is reduced, thus reducing the need to use additional complicated control methods. However, if the weeds were still present in the field, the manual weeding methods, i.e., drowning the weeds in the mud, was practiced. As for algae growth, apart from the dry and wet method, the *Bacillus subtilis* powder was used to control the leftover algae, which also formed a nitrogen source for the rice crop plants. The rice crop pest and diseases were controlled using organic pesticides (a mixture of saponin, 50% phosphorous acid, and 50% potassium hydroxide), which were sprayed on a fortnightly basis, i.e., a week after PNSB treatment was applied. Organic tea seed cake pallets (16% saponin), an extract from camellia seeds, were also spread across the field to control the snail problem at the early stage of plant growth.

### 4.4. Field Data Collection

From WAT 4, the field data such as plant height, tiller number, leaf chlorophyll content, and plant lodging resistance were collected. In each field, 10 plants were randomly selected and marked for fixed data collection every week until WAT 12 (late vegetative and early reproductive stage). Before the field data was collected, it was made sure that the field was partly dried for accurate data collection, especially for plant height and tiller number. A simple measuring tape was used to measure the plant height from the soil surface to the tip of the tallest leaf. Individual tillers were carefully counted to record the tiller number. As for the leaf chlorophyll content, the SPAD-502 chlorophyll meter (Konica Minolta, Inc., Japan) was used to determine the relative amount of chlorophyll [53] in the rice crop leaf. Six points on each leaf of selected 3 leaves from each plant were analyzed for relative chlorophyll content. The lodging resistance was determined using the YYD-IB Plant Stem Strength Tester (Wenzhou Tripod Instrument Manufacturing Co., Ltd., Wenzhou, China).

### 4.5. Rice Root Growth and Biomass Analysis

As much as above ground, below ground performances of plants are equally important. Therefore, this research evaluated below-ground performances of PNSB-treated and untreated plants, such as root length, root volume, and root dry weight. Three rice crop seedlings were transplanted into 2 transparent 60 cm long, 70 cm high, and 15 cm wide root boxes in a greenhouse, and the treatments were applied similarly to that in the field. Daily, the setup was monitored to ensure no forms of uncertainty could affect the results. On WAT 12, the plants were carefully removed from the root boxes by gently washing the soil off the roots. The plants were then packed in a bag with appropriate identification and taken to the lab for measurements. Root length was measured using a regular measuring tape, whereas root volume was determined by the water displacement technique. Then, the roots were dried in the precision oven DV-1202L (CHANNEL Qianrui Instrument Co., Ltd., Taiwan) at a temperature of 40 °C until a constant weight was achieved, and the dry weight of the roots was measured using the PB3002-S precision balance (Swiss Merchant METTLER TOLEDO Co., Ltd., Taiwan).

### 4.6. Antioxidant Enzyme Activity Analysis

Antioxidant enzyme activity such as ascorbate peroxidase (APX), catalase (CAT), glutathione reductase (GR), and superoxide dismutase (SOD) was analyzed from WAT 4 to WAT 12 weekly and determined based on the protein content of the enzyme extract using the method of Bradford [54]. The samples collected in the field were instantly kept in dry ice and transferred to the lab for analysis. During laboratory analysis, a fresh leaf sample (1.5 g) was grided using the liquid nitrogen and then homogenized with sodium phosphate buffer (50 mM; pH 6.8 for APX, CAT, GR, and 50 mM; pH 7.4 for SOD) for further grinding before it was given an ice bath. Then, the solution was centrifuged at 12,000× *g* for 20 min (APX, CAT, GR) and 15,000× *g* for 30 min (SOD) using Velocity 14R refrigerated Centrifuge (Dymamica Scientific Ltd., Livingston, UK) at 4 °C, and the supernatant was collected.

The APX activity was analyzed by the method of Nakano and Asada [55]. The absorbance was measured at 290 nm for 1 min using Double Beam U-2900 Spectrophotometer (Hitachi High-Tech Corporation, Tokyo, Japan). As the concentration of ascorbate (AsA) decreased, the absorbance at 290 nm also reduced, and the extinction coefficient of AsA (2.8 mM^−1^ cm^−1^) was used to calculate the APX activity. One unit of APX was defined as the amount of enzyme needed to degrade 1 mole of AsA in 1 min. The CAT activity was analyzed by the method of Kato and Shimizu [39]. The reduction in hydrogen peroxide amount was measured at 240 nm, and the extinction coefficient (40 mM^−1^ cm^−1^) was used to calculate CAT activity. One unit of CAT was defined as the amount of enzyme needed to degrade 1 mole of hydrogen peroxide in 1 min. The GR activity was analyzed by the method of Foster and Hess [56]. One unit of GR was defined as the amount of enzyme needed to decrease the absorbance at 340 nm at 1 min. Finally, the SOD activity was analyzed using the method by Paoletti et al. [57]. One unit of SOD was defined as the amount of enzyme that inhibited the rate of NADH oxidation by 50% in the blank sample.

### 4.7. Statistical Analysis

The data was analyzed using the International Business Machines SPSS Statistics for Windows, version 26 (International Business Machines Corporation, Armonk, NY, USA). The mean comparison was performed using the independent sample *t*-test, and results were expressed as means ± standard error. The graphs and charts were produced using Origin 2019 software (Origin Lab Corporation., Northampton, MA, USA).

## 5. Conclusions

From this experiment, it can be concluded that the foliar application of PNSB positively affects the growth and the yield of the rice crop plants cultivated under field conditions but has no effect on the antioxidant enzyme activity. Even with low light and cooler temperatures, the PNSB significantly improves the growth and yield of rice crop plants. However, since PNSB is a non-spore-forming bacteria, their overall performances under field conditions are affected [58], and therefore, further field trials are needed to understand the effectiveness of PNSB under rice and other field crop cultivation. In addition, the different strains of PNSB, application methods, frequency of applications, and application rates should also be investigated thoroughly under field conditions to make PNSB an important bacterium for sustainable agriculture and provide solutions to agronomic and physiological challenges under climate change.

## Figures and Tables

**Figure 1 plants-11-02452-f001:**
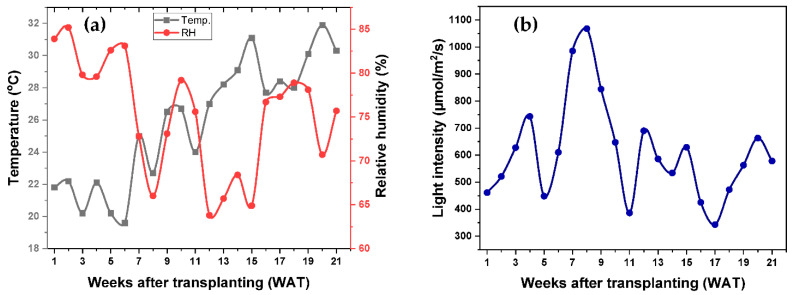
Agrometeorological data (**a**) average weekly temperature and relative humidity, and (**b**) light intensity recorded throughout the growing period.

**Figure 2 plants-11-02452-f002:**
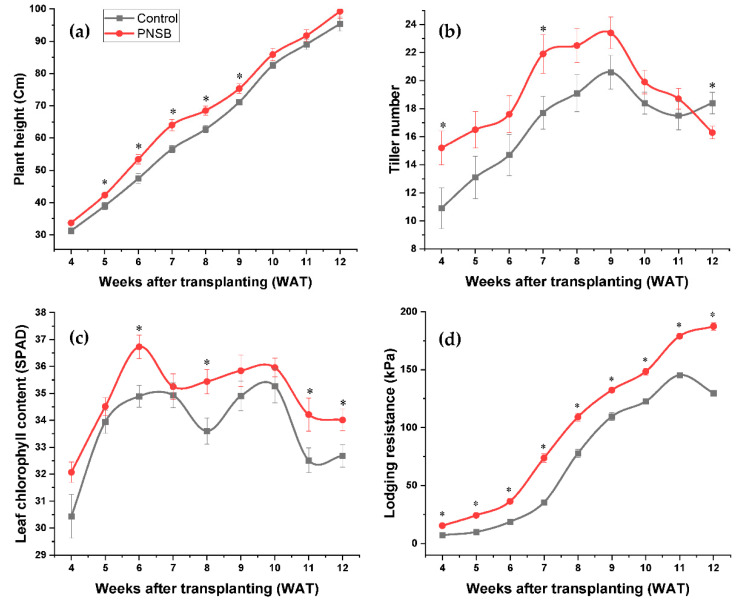
Agronomic performances (**a**) average plant height, (**b**) average tiller number, (**c**) average leaf chlorophyll content, and (**d**) average lodging resistance of rice crop under PNSB-treated and untreated plants. * denotes significant differences between control and treated plants in the respective weeks; bars represent standard error.

**Figure 3 plants-11-02452-f003:**
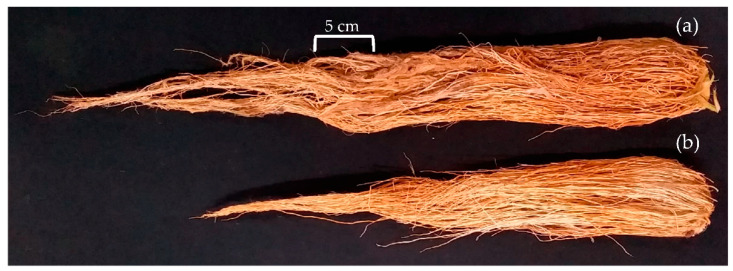
Root length of (**a**) PNSB-treated and (**b**) untreated rice crop plants investigated at the end of the vegetative stage.

**Figure 4 plants-11-02452-f004:**
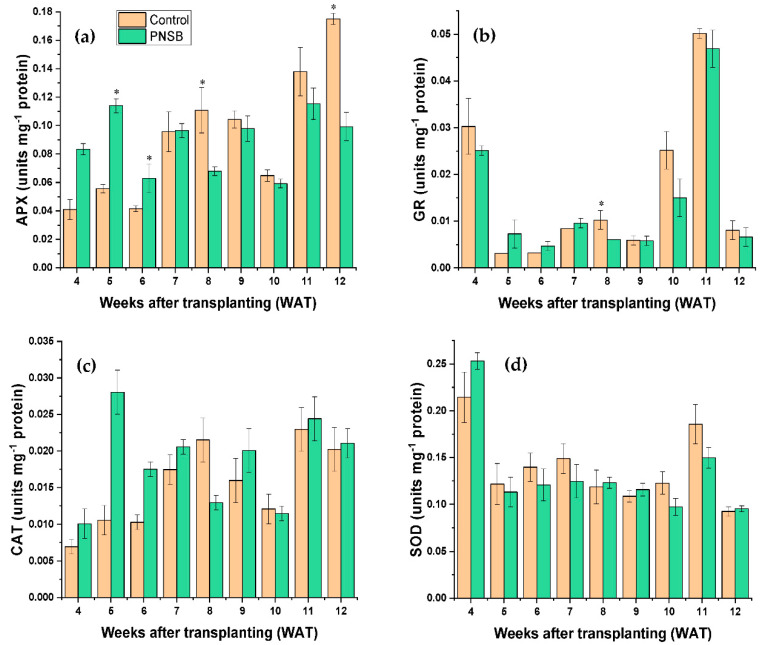
Antioxidant enzymes (**a**) ascorbate peroxidase, (**b**) glutathione reductase, (**c**) catalase, and (**d**) superoxide dismutase activities of PNSB-treated and untreated rice crop plants. * denotes significant differences between control and treated plants in the respective weeks; bars represent standard error.

**Table 1 plants-11-02452-t001:** Root length, root volume, and root dry weight of PNSB treated and untreated rice crop plants.

Treatment	Root Length (cm)	Root Volume (cm^3^)	Root Dry Weight (g)
Control	43.5 ± 0.21 ^b^	1200 ± 0.00 ^a^	18.1 ± 2.20 ^b^
PNSB	57.8 ± 2.65 ^a^	1333 ± 66.7 ^a^	41.6 ± 4.01 ^a^

Values are mean ± SE (*n* = 3). Means in the same column, followed by the same letter(s), are not significantly different (*p* ≤ 0.05).

**Table 2 plants-11-02452-t002:** Yield components of PNSB-treated and untreated rice crop plants.

Treatment	Productive Tillers/Plant (%)	Average Grains/Plant	Average Grains/Panicle	Biological Yield (t ha^−1^)
Control	48.00 ± 2.40 ^b^	1613.70 ± 55.75 ^b^	80.50 ± 3.51 ^a^	17.46 ± 1.42 ^a^
PNSB	64.44 ± 2.72 ^a^	2584.00 ± 172.32 ^a^	82.17 ± 3.22 ^a^	14.37 ± 0.90 ^a^

Values are mean ± SE (*n* = 10). Means in the same column, followed by the same letter(s), are not significantly different (*p* ≤ 0.05).

**Table 3 plants-11-02452-t003:** Yield performances of PNSB-treated and untreated rice crop plants.

Treatment	Grain Yield (t ha^−1^)	Grain Fertility (%)	1000 Grain Weight (g)	Harvest Index (%)
Control	5.42 ± 0.35 ^b^	42.02 ± 3.68 ^a^	23.98 ± 0.07 ^b^	0.33 ± 0.03 ^b^
PNSB	8.10 ± 0.73 ^a^	46.99 ± 1.81 ^a^	24.37 ± 0.11 ^a^	0.56 ± 0.04 ^a^

Values are mean ± SE (*n* = 10). Means in the same column, followed by the same letter(s), are not significantly different (*p* ≤ 0.05).

## Data Availability

Not applicable.

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
