# Peer review of "Foliar Application of *Rhodopseudomonas palustris* Enhances the Rice Crop Growth and Yield under Field Conditions"

_plants, 2022, doi:10.3390/plants11192452_

Round 1
Reviewer 1 Report
COMMENTS
INTRODUCTION SECTION
Marginal land is already highly unproductive with no agricultural value, unless, like you mentioned, there is heavy investment in terms of inputs. Line 36-37
True, rising temperature due to climate change may encourage development of new pest and pathogens, however, it may lead to extinction of others especially those that thrive best at lower temperatures. Increased C02 levels may be an advantage in terms of yield, since plants require CO2 for photosynthesis. Many parts of the world, especially temperate regions with long winters, cold falls/autumn and really hot summers, are already dealing with short planting seasons (just spring) hence, increasing temperatures may be an advantage in a sense that previously seasons too cold for planting may become warm, hence creating longer planting seasons. Some points need more than a sentence to make sense, you may need to revisit how you are putting out these issues concerning climate change. Lines 40-46
On line 57, not all microorganisms have the potential to do the things you have listed. Be specific, which category of microorganisms has the potential to do that?
Write something about Rhodopseudomonas palustris in your introduction, not every reader may be able to know that it is a Purple non-sulfur bacteria.
In general you may have to improve the introduction basing on the points above.
MATERIALS AND METHODS
You mention two blocks but do not detail how the experiment was set up on these two blocks. How many replicates? What were the controls? What was the experimental design? Was the experiment repeated? These are details you cannot afford to miss in an experiment because then it cannot be reproduced by other researchers. Pay attention to tenses. Some are in past while some are in present tense.
Re-write your materials and methods and fill in all the missing details.
RESULTS AND DISCUSION
Results are presented well. The discussion section can be improved.
OTHER COMMENTS
The grammar needs to be improved.
Author Response
Dear reviewer,
Thank you very much for your valuable comments and suggestions on our manuscript. We have made the changes to the manuscript based on your comments and suggestions. Here is a brief of what changes we have made.
Reviewer 1
INTRODUCTION SECTION
- Marginal land is already highly unproductive with no agricultural value, unless, like you mentioned, there is heavy investment in terms of inputs. Line 36-37
In lines 37 and 38, we have made changes to the sentence in the manuscript so that it correlates and makes sense with the previous sentence.
- True, rising temperature due to climate change may encourage development of new pest and pathogens, however, it may lead to extinction of others especially those that thrive best at lower temperatures.
Agreed, and we have changed the sentence in lines 49-51.
- Increased C02 levels may be an advantage in terms of yield, since plants require CO2 for photosynthesis.
The sentence has been rewritten, as shown in lines 43-48.
- Many parts of the world, especially temperate regions with long winters, cold falls/autumn and really hot summers, are already dealing with short planting seasons (just spring) hence, increasing temperatures may be an advantage in a sense that previously seasons too cold for planting may become warm, hence creating longer planting seasons.
Agreed, and we have made changes in the sentence, as shown in lines 43-48.
Some points need more than a sentence to make sense, you may need to revisit how you are putting out these issues concerning climate change. Lines 40-46- Changes were made accordingly, as mentioned above.
- On line 57, not all microorganisms have the potential to do the things you have listed. Be specific, which category of microorganisms has the potential to do that?
In line 63, an addition in the sentence specifies which category of microorganisms has those potentials.
- Write something about Rhodopseudomonas palustrisin your introduction, not every reader may be able to know that it is a Purple non-sulfur bacteria.
A brief about Rhodopseudomonas palustris was included in lines 70-76.
In general, you may have to improve the introduction basing on the points above. Done
MATERIALS AND METHODS
- You mention two blocks but do not detail how the experiment was set up on these two blocks. How many replicates? What were the controls? What was the experimental design? Was the experiment repeated? These are details you cannot afford to miss in an experiment because then it cannot be reproduced by other researchers.
Since we wanted to mimic the actual conditions farmers might face, we used two entire fields for research with 10 plants tagged for field data collection. In addition, for antioxidant enzyme activity, random plants were selected weekly for sample collection. More details can be found in lines 89-97.
- Pay attention to tenses. Some are in past while some are in present tense. Changes were made where only past tense was used throughout the manuscript for consistency.
Rewrite your materials and methods and fill in all the missing details. Done
RESULTS AND DISCUSION
Results are presented well. The discussion section can be improved. The discussion part was improved
OTHER COMMENTS
The grammar needs to be improved. The native English speaker again checked the manuscript to improve any grammatical errors.

Reviewer 2 Report
The English is poor writen and must be improve, the significative differences observed in the figures are not enough, the SE or SD must be show in each point. The figure of root inoculated and non-inoculated does represent the data of Table 1? The Figure 4 the Authors have to explain the significative differences correspond for what ? Non-inoculated and inoculated plants in which week, or for the whole treatment, in my view point is not clear. Why Figures 4C and D no differences were observed ?
Author Response
Reviewer 2
Dear reviewer,
Thank you very much for your valuable comments and suggestions on our manuscript. We have made the changes to the manuscript based on your comments and suggestions. Here is a brief of what changes we have made.
- The English is poor writing and must be improve. The native English speaker again checked the manuscript to improve any grammatical errors.
- The significative differences observed in the figures are not enough, the SE or SD must be show in each point. Respective figures were modified, and standard error bars were added.
- The Figure of root inoculated and non-inoculated does represent the data of Table 1. Figure 3 shows the root length between the inoculated and uninoculated plants, which we mentioned in line 277.
- The Figure 4 the Authors have to explain the significative differences correspond for what? Changes are made accordingly.
- Non-inoculated and inoculated plants in which week, or for the whole treatment, in my viewpoint is not clear. In lines 118 -120, the sentence indicates that the treatment was applied from 4 weeks after transplanting until the early reproductive stage. And as shown in Figure, treatment was applied from 4 weeks after transplanting to 12 after transplanting.
- Why Figures 4C and D no differences were observed? In lines 491-497, the reason has been added as to why no significant differences in antioxidant enzyme activity were observed between PNSB treated and untreated plants.

Round 2
Reviewer 1 Report
You mention that one block was used for treatments while the other for control. Are you certain that your treatments are the only significant variation between the two blocks? i.e. there no variations between these blocks that may have a significant effect on your results, not caused by the treatments? How close to each other are these blocks. Perhaps you need to mention this in your methods.
Author Response
Dear reviewer,
Thank you very much for your valuable comments and suggestions on our manuscript. We have made the changes to the manuscript based on your comments and suggestions, as shown below.
Comment
You mention that one block was used for treatments while the other for control. Are you certain that your treatments are the only significant variation between the two blocks? i.e. there no variations between these blocks that may have a significant effect on your results, not caused by the treatments? How close to each other are these blocks. Perhaps you need to mention this in your methods.
We have added the sentence below in lines 93-96 to answer this part.
“The blocks were set up parallel to each other in the same area under similar environmental conditions. All management practices, from land preparation to harvesting, were carried out similarly in each block to avoid biases in the results”.
Reviewer 2 Report
The changes improved the manuscript quality.
Author Response
Dear reviewer,
Thank you very much for your valuable comments and suggestions on our manuscript.